# Cardiac Cx43 Signaling Is Enhanced and TGF-β1/SMAD2/3 Suppressed in Response to Cold Acclimation and Modulated by Thyroid Status in Hairless SHR^M^

**DOI:** 10.3390/biomedicines10071707

**Published:** 2022-07-14

**Authors:** Katarina Andelova, Barbara Szeiffova Bacova, Matus Sykora, Stanislav Pavelka, Hana Rauchova, Narcis Tribulova

**Affiliations:** 1Centre of Experimental Medicine, v.v.i., Slovak Academy of Sciences, 84104 Bratislava, Slovakia Republic; katarina.andelova@savba.sk (K.A.); barbara.bacova@savba.sk (B.S.B.); matus.sykora@savba.sk (M.S.); 2Institute of Molecular Genetics, v.v.i., Academy of Sciences of the Czech Republic, 14220 Prague, Czech Republic; stanislav.pavelka@img.cas.cz; 3Institute of Physiology, v.v.i., Academy of Sciences of the Czech Republic, 14220 Prague, Czech Republic; hana.rauchova@fgu.cas.cz

**Keywords:** hairless SHR^M^, cold acclimation, cardiac Cx43, extracellular matrix, thyroid hormones

## Abstract

The hearts of spontaneously hypertensive rats (SHR) are prone to malignant arrhythmias, mainly due to disorders of electrical coupling protein Cx43 and the extracellular matrix. Cold acclimation may induce cardio-protection, but the underlying mechanisms remain to be elucidated. We aimed to explore whether the adaptation of 9-month-old hairless SHR^M^ to cold impacts the fundamental cardiac pro-arrhythmia factors, as well as the response to the thyroid status. There were no significant differences in the registered biometric, redox and blood lipids parameters between hairless (SHR^M^) and wild type SHR. Prominent findings revealed that myocardial Cx43 and its variant phosphorylated at serine 368 were increased, while an abnormal cardiomyocyte Cx43 distribution was attenuated in hairless SHR^M^ vs. wild type SHR males and females. Moreover, the level of β-catenin, ensuring mechanoelectrical coupling, was increased as well, while extracellular matrix collagen-1 and hydroxyproline were lower and the TGF-β1 and SMAD2/3 pathway was suppressed in hairless SHR^M^ males compared to the wild type strain. Of interest, the extracellular matrix remodeling was less pronounced in females of both hypertensive strains. There were no apparent differences in response to the hypothyroid or hyperthyroid status between SHR strains concerning the examined markers. Our findings imply that hairless SHR^M^ benefit from cold acclimation due to the attenuation of the hypertension-induced adverse downregulation of Cx43 and upregulation of extracellular matrix proteins.

## 1. Introduction

The regulation of body temperature is fundamental to the maintenance of homeostasis, and adaptive thermogenesis, tightly regulated by the central nervous system, is essential for survival [1,2]. Catecholamines and thyroid hormones (TH) are the most important factors impacting thermogenesis. TH activate thermogenesis in brown adipose tissue, and white adipose tissue can undergo browning via adrenergic stimulation [3]. The adaptation to cold results in the deiodination of thyroxine (T_4_) and causes an increase of the triiodothyronine (T_3_) levels in blood. Higher T_3_ leads to an increase in expression of uncoupling protein 2, following an increase in heat production [4,5]. 

Cardiovascular system responses to thermoregulatory challenges and cold acclimation may induce cardio-protection [6,7]. However, there is still a gap in the knowledge on the underlying mechanisms by which a temperature challenge or adaptation to cold induces cardiac responses. Moreover, molecular mechanisms linked with cardio-protection to maintain heart function in response to cold adaptation are poorly elucidated as well. 

The heart is an electromechanical pump, and gap junction channels formed by connexin-43 (Cx43) ensure coupling between cardiac myocytes to enable the transmission of electrical and molecular signals, resulting in coordinated contractions [8,9], while the downregulation of Cx43 and/or channel dysfunction, as well as abnormal Cx43 topology, jeopardize synchronous cardiac electromechanical functions and render the heart prone to developing arrhythmias, including potentially lethal ventricular fibrillation (VF) [10,11]. Moreover, reduced inter-myocyte coupling and communication activate pro-fibrotic signaling, followed by excessive extracellular matrix (ECM) deposition [12], that profoundly contribute to arrhythmogenesis and heart mechanical failure [13]. 

Of interest, an increased myocardial expression of Cx43 was demonstrated in hibernators during cold acclimatization and hibernation [14]. The upregulation of Cx43 appears to be a mechanism by which the hibernators avoid lethal arrhythmia during hibernation and arousal [15], despite marked changes in body temperature, which, in non-hibernating subjects and humans, induce VF. The questions arise as to whether the upregulation of cardiac Cx43 in response to cold adaptation is a general phenomenon occurring in non-hibernating animals and whether there are sex-related differences.

Genetically, hairless strains of rodents represent rewarding models to study adaptive thermogenesis. It has been reported that hairless coisogenic spontaneously hypertensive rats (SHR^M^), harboring the mutant desmoglein-4 gene, exhibit increased thermogenesis due to a metabolic adaptation to cold [16]. The standard ambient temperature of 22 °C lies well below thermoneutrality for such rats due to the diminished insulating capacity. However, the response of the heart to cold acclimation was not investigated in this strain. It would be of a great interest, since hypertension increases the arrhythmia risk, and SHR has been shown as prone to VF due to the downregulation of myocardial Cx43, its abnormal topology and ECM alterations (fibrosis) [13]. Moreover, SHR differ in Cx43 and ECM responses to altered the thyroid status when compared to normotensive rats [17].

Taken together, we aimed to investigate whether there are specific arrhythmia-related differences between the hearts of hairless vs. wild type SHR and in the response to the induced hyperthyroid or hypothyroid status. In this context, our main objective was to explore the expression of myocardial Cx43 and its topology, as well as the key markers of fibrogenesis in hairless SHR^M^ males and females. Additionally, the expression levels of the selected proteins that may modulate Cx43 channel’s function were examined in the heart of both SHR strains.

## 2. Materials and Methods

### 2.1. Animals and Experimental Design

The experiments were performed on 9-month-old male and female hairless coisogenic spontaneously hypertensive rats (SHR) harboring the mutant desmoglein 4 (Dsg4) gene (SHR^M^), as well as age and sex-matched wild type SHR. Besides, male and female Wistar Kyoto rats (WKY) were used as a reference normotensive strain. Animals were obtained from the Institute of Physiology, v.v.i., Academy of Sciences of the Czech Republic, Prague. The Dsg4 mutation was originally found in the SHR.BN-chr.1 congenic strain and was transferred on the SHR genetic background by backcross breeding [16]. The animals were housed at 22 °C with 12-h light/dark cycles and ad libitum access to tap water and standard laboratory chow. Experiments were performed in agreement with the Animal Protection Laws of the Czech Republic. The maintenance and handling of the animals were performed in accordance with the “Guide for the Care and Use of Laboratory Animals” published by the U.S. National Institutes of Health (NIH Publication, 8th ed., revised 2011) and approved by the Ethics Committee of the Institute of Physiology v.v.i, Academy of Sciences of the Czech Republic, Prague.

Male and female SHR were randomly divided into 6 experimental groups: male SHR (n = 6), female SHR (n = 6), male hyperthyroid SHR TH (n = 5), female hyperthyroid SHR TH (n = 5), male hypothyroid SHR HY (n = 5) and female hypothyroid SHR HY (n = 5). Male and female SHR^M^ were randomly divided into 6 experimental groups: male SHR^M^ (n = 6), female SHR^M^ (n = 6), male hyperthyroid SHR^M^ TH (n = 5), female hyperthyroid SHR^M^ TH (n = 5), male hypothyroid SHR^M^ HY (n = 5) and female hypothyroid SHR^M^ HY (n = 5). The hyperthyroid status was established by the intraperitoneal injection of 3,3′,5-triiodo-L-thyronine (T_3_; Sigma Aldrich, St. Louis, MO, USA) at 0.15 mg/kg b.w. three times weekly, and the hypothyroid status was induced by a 0.05% solution of methimazole (Sigma Aldrich, St. Louis, MO, USA) in the drinking water. Wistar Kyoto (male WKY, n = 5; female WKY, n = 5) nontreated normotensive rats were used as the control reference strain. At the end of the experiment, the animals were euthanized with 100 mg/kg b.w. of ketamine (Narketan; Vetoquinol UK Ltd., Towcester, UK), followed by 10 mg/kg b.w. of myorelaxant xylazine (Xylapan; Vetoquinol UK Ltd., Towcester, UK), and the body weight was registered. Then, the chest was opened, and the heart was quickly excised into ice-cold saline, followed by weight registration and heart tissue sampling. Blood samples were taken from the thoracic aorta. All samples were stored in a freezer at −80 °C. 

### 2.2. Biometric, Blood Samples and Cardiac Left Ventricular Tissue Parameters Monitoring 

Body weight, heart and left ventricular weight of rats were registered at the end of the experiment. The systemic blood pressure (BP) was monitored via the cannulated carotid artery, as previously described [18], in WKY, SHR and SHR^M^ males and females. Postprandial levels of triglycerides (TG), total cholesterol (TC) and high-density lipoproteins (HDL) were assessed using available kits (Pliva-Lachema Diagnostika, Brno, Czech Republic). Low-density lipoprotein-cholesterol (LDL) was estimated indirectly using the concentration relations: LDL = TC − (TG/5) − HDL. Concentrations of the total L-thyroxine (tT_4_) and 3,3′,5-triiodo-L-thyronine (tT_3_) in blood sera of the rats were determined by radioimmunoassay (RIA) using commercial RIA kits with [^125^I]-T_4_ and [^125^I]-T_3_ tracers (Immunotech/Beckman Coulter Co., Prague, Czech Republic), as described previously [19]. The left ventricular thiobarbituric acid reactive substances (TBARS) were determined as described previously [17], and the content of reduced glutathione (GSH) was assessed in the left ventricular tissue, as described previously [20].

### 2.3. Myocardial Histology and Enzyme Histochemistry

Conventional hematoxylin–eosin and Van Gieson staining of heart left ventricular tissue sections were used for a light microscopic examination of the myocardial structure (Zeiss Apotome 2 microscope: Carl Zeiss, Jena, Germany). Catalytic enzyme histochemistry was performed according to [21] using 10-μm-thick heart cryostat sections to demonstrate the activities of capillary endothelium-related alkaline phosphatase (AP, E.C.3.1.3.1) with naphthol AS-MX phosphate as a substrate and dipeptidyl peptidase-4 (DPP4, E.C.3.4.15.4) with glycyl-L-proline-4-methoxy-beta naphthylamide as a substrate to detect functional arteriolar and venular capillary network. For quantification of intensity of AP and DPP4 histochemical reactions (corresponding to the enzyme activity), randomly selected areas (15 per heart) of positive signal were analyzed and defined as the number of pixels with a code lower than 128 on the “0–255 RGB color scale”. The total number of positive pixels was expressed as a total integral optical density per area (IOD) (Image-Pro Plus). 

### 2.4. Immunofluorescence Labelling of Myocardial Cx43, Cadherin, β-Catenin and Quantitative Image Analysis

Cryostat sections from the left ventricle were used for the in situ immunodetection of Cx43, cadherin and β-catenin. The tissue sections were fixed in ice-cold methanol, permeabilized in 0.3% Triton X-100, washed in phosphate-buffered saline (PBS) and blocked with 1% bovine serum. The sections were incubated overnight with primary anti-Cx43 antibody (1:500, MAB3068, CHEMICON International, Inc., Temecula, CA, USA), anti-cadherin (1:300, sc-7939, Santa Cruz, Dallas, TX, USA) and with anti-β-catenin (1:250, sc-7963, Santa Cruz, Dallas, TX, USA) at 4 °C, washed in PBS and subsequently incubated for one and a half hours with secondary antibodies conjugated with anti-mouse FITC-fluorescein isothiocyanate (1:500, Jackson Immuno Research Labs, West Grove, PA, USA) or with anti-rabbit Alexa Fluor 594 (1:500, Jackson Immuno Research Laboratory Labs, West Grove, PA, USA). 

The immunostaining of Cx43, cadherin and β-catenin was examined using a Zeiss Apotome 2 microscope (Carl Zeiss, Jena, Germany), and digital images were used for the quantitative analysis (Image-Pro Plus). Ten randomly selected areas were examined per heart. The quantification of Cx43 located on lateral sides of the cardiomyocytes (as a marker of the pathophysiological topology of Cx43) was performed as previously described [22]. After manual delineation of terminal intercalated disc-related Cx43 immunolabeling, the difference between the total IOD and IOD of terminal Cx43 corresponded to lateral Cx43. It was expressed as a percentage calculated from the ratio of lateral topology divided by the total IOD.

### 2.5. Determination of Collagen Content by Hydroxyproline Measurement

The hydroxyproline content in the left ventricle tissue was estimated by the spectrophotometric method, as described previously [23]. Samples were hydrolyzed in 6 M HCl for 3 h at 130 °C. Dried samples were treated at room temperature with chloramine T in the acetate–citrate buffer (pH 6.0), and the reaction was stopped after 20 min by adding Ehrlich’s reagent solution. It was followed by incubation at 65 °C for 15 min. The concentration of hydroxyproline was measured spectrophotometrically at 550 nm and expressed in mg per total weight of the left ventricle [22].

### 2.6. Determination of Myocardial Protein Levels by Western Blotting

Frozen left ventricular tissue was powdered in liquid nitrogen and homogenized in SB20 lysis buffer (20% SDS, 10 mmol/L EDTA and 100 mmol/L Tris, pH 6.8). To analyze the active form of collagen-1, the samples were prepared in Laemmli buffer without 2-mercaptoethanol and loaded onto gels without denaturation. For other proteins, the tissue lysate was diluted in the Laemmli sample buffer and boiled for 5 min, and an equal amount of protein was loaded in each well, followed by separation on SDS-PAGE 10% bis-acrylamide gels at a constant voltage of 120 V (Mini-Protean TetraCell, Bio-Rad, Hercules, CA, USA), as previously described [17]. Subsequently, the proteins were transferred to a nitrocellulose membrane (0.2-μm pore size, Advantec, Tokyo, Japan) and blocked for 4 h with 5% fat-free milk in Tris-buffered saline containing 0.1% Tween 20 (TBST). The membrane was incubated overnight with the primary antibodies listed in Table 1. The membrane was subsequently washed in TBST and incubated for 1 h with a horseradish peroxidase-linked secondary anti-rabbit antibody (1:2000, 7074S, Cell Signaling Technology, Denver, CO, USA) and anti-mouse antibody (1:2000, 7076C, Cell Signaling Technology, Denver, CO, USA). The enhanced luminol-based chemiluminescent was used for visualization of the proteins and the quantification of the relevant bands was assessed densitometrically using Carestream Molecular Imaging Software (version 5.0, Carestream Health, New Haven, CT, USA) and normalized to GAPDH.

### 2.7. Statistical Evaluation

Differences between groups were evaluated using one-way analysis of variance (ANOVA) and Bonferroni’s multiple comparison test. The Kolmogorov–Smirnov normality test was used to examine whether variables are normally distributed. Data were expressed as means ± standard deviations (SD); *p* < 0.05 was considered to be statistically significant.

## 3. Results

### 3.1. Biometric, Blood Samples and Cardiac Left Ventricular Tissue Parameters of Experimental Rats

Comparing to the systolic blood pressure in normotensive WKY rats (114.2 ± 17 mmHg in males and 99.5 ± 11 mmHg in females), this parameter was significantly elevated in either sex of the wild type SHR (190.0 ± 7 mmHg in males and 179.6 ± 12 mmHg in females) and in hairless SHR^M^ (182.9 ± 6 mmHg in males and 169.7 ± 9 mmHg in females). As shown in Table 2, wild type SHR males exhibited higher body weights when compared to SHR^M^, whereas there was no difference in body weights between these two strains in the females. The heart and left ventricular weights did not noticeably differ between SHR and SHR^M^, regardless of the sex, but both cardiac parameters were significantly higher when compared to normotensive WKY. Hyperthyroidism resulted in a decrease in body weight in SHR males and, to a lesser extent, in SHR^M^ males but not in females. There were no significant differences in the myocardial TBARS levels between wild type SHR and hairless SHR^M^ or in comparing to WKY rats, but there was a tendency to increase due to hyperthyroidism in both SHR strains. The myocardial levels of GSH did not differ significantly among the experimental groups, regardless of the strain and sex.

As shown in Table 2, the serum total T_3_ and T_4_ concentrations were similar in wild type SHR and hairless SHR^M^ males, but in SHR females, the T_3_ concentrations were a bit higher in comparison with SHR^M^ females. While comparing to WKY rats, the serum T_3_ concentrations were significantly higher in both SHR strains, regardless of the sex. The serum triglycerides, total cholesterol, HDL and LDL cholesterol did not significantly differ in SHR vs. SHR^M^, but these parameters were lower in both SHR strains compared to WKY rats. There were no significant differences in the atherogenic index, expressed as the TC/HDL ratio, among the groups.

### 3.2. Myocardial Histology and Capillary Enzyme Histochemistry

Hematoxylin–eosin staining (Figure 1) revealed that, compared to WKY rats, the left ventricular tissue of both wild type SHR and SHR^M^ of either sex exhibited focal areas infiltrated with polymorphonuclears. This histopathological feature persisted regardless of the hyper- or hypothyroid status. Besides, Van Gieson staining for collagen deposition (Figure 2) revealed interstitial and perivascular fibrosis in both SHR strains males and females but to lesser extent in SHR^M^. Of note, the representative microscopic images indicate the most pronounced area of fibrosis detected in each experimental group of rats. The hypothyroid status enhanced the myocardial fibrosis in both SHR strains. 

Histochemical determination of the activity of alkaline phosphatase (AP) that points out the function and myocardial density of the arterial part of capillaries is demonstrated in Figure 3. Alterations of the capillary network AP activity may reflect the myocardial adaptation to hypertension to maintain the function of a structurally remodeled and fibrotic heart. Compared to normotensive WKY rats, the AP activity was significantly increased in both wild type SHR and SHR^M^, regardless of the sex, and was enhanced due to hypothyroidism in males and females of both SHR strains, as assessed by a quantitative image analysis.

The activity of dipeptidyl peptidase-4 (DPP4), that points out the function and density of the venous part of capillaries, is demonstrated in Figure 4. Enhanced DPP4 activity is considered detrimental in pathophysiological conditions due to the implication in collagen metabolism and proinflammatory signaling. A quantitative image analysis revealed that the DPP4 activity was significantly lower in SHR^M^ regardless of the sex when compared to normotensive WKY rats or wild type SHR. Hyperthyroidism resulted in a decrease of DPP4 activity, while an increase was observed in hypothyroidism in both wild type SHR and SHR^M^, regardless of the sex.

### 3.3. Myocardial Protein Levels of Cx43 and Its Variants

The expression level of the Cx43 protein was significantly reduced in the left ventricle of SHR males and females, as well as in SHR^M^ males (Figure 5), compared to normotensive WKY rats. However, compared to wild type SHR males and females, the protein levels of Cx43 were significantly higher in sex-matched SHR^M^. The hypothyroid status enhanced the Cx43 levels in wild type SHR, as well as in SHR^M^, and the increase was more pronounced in females than in males of both SHR strains. The hyperthyroid status resulted in a decrease of Cx43 in SHR^M^ males while not in wild type SHR males, but an increase was demonstrated in SHR females and. to a lesser extent, in SHR^M^ females.

The levels of expression of the Cx43 variant phosphorylated at serine368 (pCx43^368^), which, is associated with a reduced channel conductivity and the Cx43 variant phosphorylated at serine279 (pCx43^279^) that hampers channel conductivity are shown in Figure 6. Compared to WKY rats, the expression of pCx43^368^ was significantly decreased in wild type SHR males and females but increased in SHR^M^, apparently in males. The hypothyroid status enhanced the pCx43^368^ expression only in wild type SHR, regardless of the sex, while not in SHR^M^. On the contrary, the hyperthyroid status reduced the pCx43^368^ protein levels only in SHR^M^ males. There were no significant changes in the pCx43^279^ protein abundance either in wild type SHR or SHR^M^, regardless of the sex, when compared to normotensive WKY rats. Both the hyperthyroid and hypothyroid status reduced the pCx43^279^ protein levels significantly in SHR^M^ males but not in females. 

### 3.4. Myocardial Topology of Cx43 and Quantification of Its Abnormal Distribution

The cardiomyocyte distribution of Cx43 in experimental rats is demonstrated in Figure 7. The obvious prevalent distribution of Cx43 is detected at the gap junction plaques of the intercalated discs in normotensive WKY rats. Besides this obvious end-to-end pattern, there was enhanced immunofluorescence labeling of Cx43 at the lateral sides of the cardiomyocytes (side-to-side pattern) in both SHR strains, regardless of the sex and thyroid status. A quantitative image analysis of Cx43-positive labeling revealed a significant increase of Cx43 at the lateral sides of the cardiomyocytes in wild type SHR males and females but to a lesser extent in SHR^M^, regardless of the sex. Moreover, there was a tendency to increase the lateral distribution of Cx43 due to the hyperthyroid status, apparently in females of both SHR strains. In the context of lateral Cx43 distribution, the use of Triton X-100 soluble and Triton X-100 insoluble portions analysis [24] could provide information about the integrity of the gap junction plaques. 

### 3.5. Myocardial Protein Levels and Topology of Cx43 Interacting Protein, β-Catenin

β-catenin is an adhesive junction protein interacting with Cx43 and impacting channel assembly and function [25]. Immunolabeling of β-catenin is confined to the intercalated disc junctions (end-to-end pattern), as shown in Figure 8. There was no difference in the topology of β-catenin among the experimental groups of rats. While the Western blotting analysis revealed (Figure 8) that, compared to WKY rats, the protein levels of β-catenin exhibited a tendency to be lower in wild type SHR but higher in SHR^M^ (significantly in females). The hyperthyroid status increased the β-catenin protein levels in wild type SHR males and females but not in SHR^M^. The hypothyroid status exhibited a tendency to decrease in the β-catenin protein levels in males while not in females of both SHR strains. Nevertheless, regarding the Cx43 function, it would be useful to explore active vs. inactive forms of β-catenin.

### 3.6. Myocardial Expression of Cx43 Interacting Protein Kinases

Cx43 interacting protein kinases by the phosphorylation of the Cx43 impact channel’s function and inter-myocyte communication [26,27]. Accordingly, PKCε, PKG and MAPK42/44 attenuate, while PKA and Akt kinase facilitate the Cx43 channel conductivity. The expression levels of the assessed protein kinases are demonstrated in Figure 9. Compared to wild type SHR, the protein levels of PKCε were increased in SHR^M^ males, as well as due to the hypothyroid status in both SHR^M^ and wild type SHR, regardless of the sex, while the expression of PKG and MAPK42/44 was not significantly altered in SHR^M^ males vs. wild type SHR or in response to the altered thyroid status. However, the MAPK42/44 expression was increased in SHR^M^ females compared to wild type SHR. The expression of PKA was lower in SHR^M^ males vs. wild type SHR but enhanced due to the hypothyroid or hyperthyroid status. There were no changes in PKA expression among the experimental rat groups in the females. Moreover, there was no significant difference in the expression of Akt kinase between the SHR^M^ and wild type SHR groups, while an increase of Akt kinase protein due to the hypothyroid status was detected in males but not in females of both SHR strains. Demonstrated changes of the protein kinase expression may have an impact on Cx43 phosphorylation and function. Therefore, their interaction with Cx43 (coimmunoprecipitation), along with its phosphorylated status, is challenging to explore.

### 3.7. Myocardial Level of Profibrotic Markers TGF-β1, SMAD2/3, Collagen-1 and Hydroxyproline

As shown in Figure 10, the protein expression of TGF-β1 was significantly higher in wild type SHR males but not in SHR^M^, compared to normotensive WKY rats. While TGF-β1 expression was not altered in females of both SHR strains when compared to WKY rats. A significant elevation of TGF-β1 was observed only in SHR^M^ females in response to the hypothyroid status. 

In parallel, there was an increase of SMAD2/3 expression in wild type SHR (significantly in males) but not in SHR^M^, regardless of the sex, when compared to WKY rats (Figure 10). The hypothyroid status increased the SMAD2/3 expression in wild type males (SHR), while the hyperthyroid status decreased it in SHR^M^. 

Moreover, there was a significant increase of collagen-1 (Figure 10) in wild type SHR males, as well as females, but not so much in SHR^M^ when compared to WKY rats. The hypothyroid status highly increased the content of collagen-1 in wild type SHR males and females but much less in SHR^M^.

The hydroxyproline levels (Figure 10) were significantly increased in wild type SHR males but not in SHR^M^ when compared to WKY rats. However, the hypothyroid status highly increased the hydroxyproline content in both SHR strains, regardless of the sex.

### 3.8. Myocardial Protein Levels of β1, β2 and β3-Adrenergic Receptors (AR)

The adrenergic stimulation is important for cold acclimation-elicited non-shivering thermogenesis and the cardiac-specific overexpression of β3-AR hampers hypertrophy and myocardial fibrosis [28,29]. According to our analyses (Figure 11), the expression of cardiac-dominant β1-AR was significantly decreased in wild type SHR, as well as in SHR^M^ males and females, when compared to normotensive WKY rats. There was a tendency to increase the expression of β1-AR in response to the hypothyroid status in wild type SHR, as well as in SHR^M^ males. Like β1-AR, the expression of β2-AR was decreased in males regardless of the SHR strain when compared to WKY rats, while the hyperthyroid status resulted in an increase of β2-AR expression in wild type SHR, as well as in SHR^M^ males. Compared to WKY rats, there were no differences in β3-AR expression among the males, unlike hairless SHR^M^ females that exhibited an increased expression of β3-AR. 

## 4. Discussion

In the current study, we characterized hairless SHR (SHR^M^) males and females adapted to cold and compared them with sex- and age-matched wild type SHR. While WKY rats were used as a reference normotensive strain to hypertensive rats, there were no significant differences in the biometric, registered blood samples and cardiac tissue parameters between the hairless SHR^M^ and wild type SHR. However, compared to WKY rats, SHR exhibited significantly lower triglycerides and cholesterol but higher circulating T_3_, as well as higher heart and left ventricular weight, likewise previously reported [17].

Of interest, the enzyme histochemistry revealed an increase of alkaline phosphatase (AP) activity in the arterial portion of the heart capillary network in males and females of either SHR strain but more pronounced in hairless SHR^M^ males when compared to WKY rats. Vascular AP is involved in purinergic metabolism and the production of adenosine, which is a potent vasodilator and anti-aggregator. However, enhanced purinergic signaling may be implicated in the adverse modulation of myocardial Cx43 channel functions in a pathological setting [9]. In contrast to AP, the activity of dipeptidyl peptidase-4 (DPP4), confined to the venous part of heart capillaries, was significantly reduced in hairless SHR^M^, regardless of the sex, but not in wild type SHR when compared to WKY rat hearts. Vascular DPP4 is a transmembrane serine protease that degrades vasoactive peptides and hormones and is also involved in collagen metabolism [30] and proinflammatory processes [31], while the inflammation deteriorates the Cx43 channel function [32]. Lower DPP4 activity attenuated capillary rarefaction and myocardial inflammation and contributed to a favorable metabolism in an ischemic heart [33]. Taken together, it appears that hairless SHR^M^ may benefit comparing to wild type SHR.

Besides the apparent hypertrophy, the left ventricular tissue of males and females of both SHR strains was focally infiltrated with polymorphonuclears and exhibited perivascular and interstitial fibrosis. Such structural remodeling underlies the arrhythmogenic substrate that increased the propensity of the heart toward VF [13,22,34]. In this context, it should be emphasized that, compared to wild type SHR, the assessed extracellular matrix proteins collagen-1 and hydroxyproline were significantly lower in hairless SHR^M^ males. In parallel, the markers of profibrotic pathways TGF-β1 and SMAD2/3 were lower as well. Unlike SHR males, the females of both SHR strains did not exhibit significant alterations of hydroxyproline, TGF-β1 and SMAD2/3 when compared to WKY rats. Our findings imply that ECM remodeling is less pronounced in hairless SHR^M^ males, as well as in females of both SHR strains, when compared to sex-matched WKY rat hearts. Altogether, it appears that the hearts of hairless SHR^M^ males and females might be less prone to developing malignant arrhythmias.

In addition to the arrhythmogenic substrate, one of the key factors impacting cardiac arrhythmias occurrence is disorders of Cx43, as previously reviewed [8,9,11]. Structural remodeling is linked with the abnormal and proarrhythmic topology of Cx43, resulting from the redistribution of Cx43 from the intercalated disc to the lateral sides of the hypertrophied cardiomyocytes, as shown previously [17], and in the current study as well. Indeed, compared to the WKY rat heart, “the lateralization” was significantly increased in SHR males but to lesser extent in hairless SHR^M^, as well as in females of either SHR strain. Apart from this, the levels of the total Cx43 protein and its variant phosphorylated at serine368 (that reduces Cx43 channel conduction and permeability [35]) were significantly decreased in SHR vs. WKY rat hearts (as was also previously demonstrated [17]) but not in hairless SHR^M^ males and females. Moreover, the expression of the Cx43 variant phosphorylated at serine279, that also hampers Cx43 channel communication [26], exhibited a tendency to decrease in hairless SHR^M^ males and females when compared to WKY rat hearts, while no changes were observed in wild type SHR. Finally, a myocardial abundance of β-catenin that impacted the Cx43 channel function [25] was increased in hairless SHR^M^ males and females in contrast to reduced levels in wild type SHR when compared to WKY rat hearts. Both β-catenin and plakoglobin have been indispensable for maintaining mechanoelectrical coupling in the heart to prevent cardiac arrhythmias [36]. Altogether, the myocardial Cx43-related findings indicate that hairless SHR^M^ might be less prone to developing malignant arrhythmias than wild type SHR, namely in males. It is important, because we have previously shown [37] that the threshold to induce ventricular fibrillation was significantly lower in wild type SHR males comparing to normotensive Wistar rats, while females, regardless the strain, are less prone to malignant arrhythmias, in part due to a higher level of Cx43 expression most likely induced by estrogen [38].

Considering the fact that the phosphorylation of Cx43 is required for Cx43 channel function, we were interested to find whether the myocardial expression of most relevant protein kinases interacting with Cx43 [26,27] differ between SHR rat strains. Indeed, the expression of PKCε was increased in hairless SHR^M^ males and MAPK42/44 in SHR^M^ females vs. sex-matched wild type SHR, while the expression of PKA was lower in hairless SHR^M^ males vs. wild type SHR. These differences are challenging for further research, since the overexpression of protein kinases may increase Cx43 phosphorylation and vice versa, thereby impacting the channel function [39].

Adrenergic stimulation is important for the cold acclimation [28,29], and adrenoceptors are directly involved in Cx43 mediated inter-myocyte communication [40]. An increase of β2-AR and β3-AR in normotensive rats exposed to cold acclimation at 8 °C for 5 weeks was reported [7]. Of interest, we revealed a distinct expression of β-adrenoceptors in the SHR vs. WKY strains. There was a decrease of cardiac dominant β1-AR protein in males, as well as females, and a decrease of β2-AR protein in males of both SHR strains when compared to WKY rats. It suggests the possible downregulation of these receptors and, hence, the attenuation of their undesirable impact on Cx43 in both SHR strains. There was no difference in β3-AR expression among WKY, wild type SHR and hairless SHR^M^ males, unlike hairless SHR^M^ females, that exhibited an increased expression of β3-AR comparing to WKY or wild type SHR. Increased β3-AR may be beneficial due its anti-fibrotic and anti-hypertrophic signaling [29].

Taken into consideration another hormonal factor impacting thermogenesis [3], such as thyroid hormones, we were interested in registering potential differences in the responses of hairless SHR^M^ vs. wild type SHR to alter the thyroid status. The latter is known to affect the propensity of the heart to arrhythmias, in part due to modulation of the myocardial expression of Cx43 [41].

We demonstrated that the expression of Cx43 and its variant phosphorylated at serine368, as well as PKCε, were increased in hairless SHR^M^; similar to wild type SHR males and females in response to their hypothyroid status. It might contribute to a lower susceptibility of the hypothyroid SHR heart to malignant arrhythmias, as it was reported in hypothyroid normotensive rats [42]. On the other hand, an undesirable response to the hypothyroid status was an increase of the myocardial matrix proteins, collagen and hydroxyproline in males of both SHR strains and, to a lesser extent, in SHR^M^ females. Of interest, the hyperthyroid status did not affect the markers of the extracellular matrix in either SHR strain but resulted in the suppression of Cx43 and its serine368 variant in hairless SHR^M^ males, as well as in wild type SHR males. It may predict a higher susceptibility to arrhythmias due to hyperthyroidism, as reported in normotensive rats and humans [41,42].

## 5. Conclusions

The results of this study indicate differences in the levels of the cardiac pro-arrhythmia factors between the hearts of hairless (in particular, in males) vs. wild type SHR while not in the response to an induced hyperthyroid or hypothyroid status. The main findings, i.e., upregulation of myocardial Cx43, along with the suppression of fibrotic factors, suggests that hairless SHR^M^ are more likely to benefit from adaption to the cold most due to suppression of a malignant arrhythmia risk. Increased thermogenesis induced by the cold might be a possible cause for observed cardiac differences in the SHR^M^ and wild type SHR. The findings are a challenge to explore the mechanisms by which adaptation to cold triggers a cardiac response.

### Limitations

We did not assess either the function of the hairless SHR^M^ heart or its susceptibility to malignant arrhythmias. This should be explored in the next study to provide the evidence that the adaptation of hypertensive rats to the cold may attenuate their higher susceptibility to lethal cardiac arrhythmias. It might be challenging for clinical trials and the innovative management of hypertensive patients. 

## Figures and Tables

**Figure 1 biomedicines-10-01707-f001:**
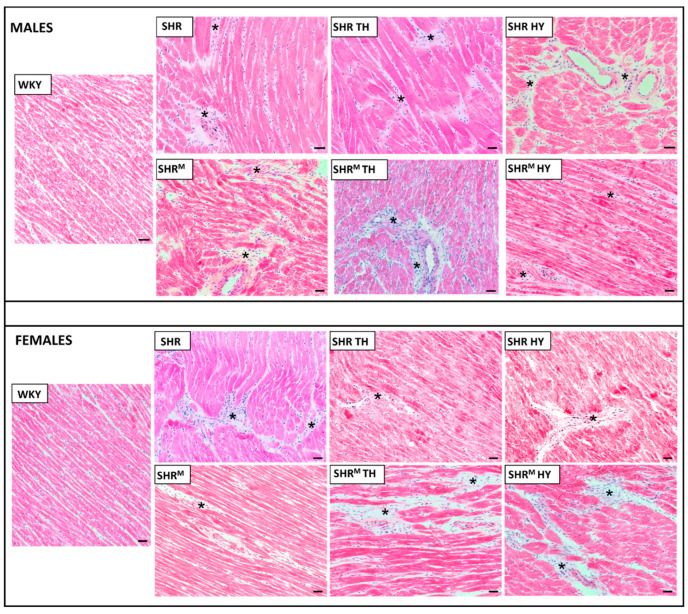
Hematoxylin–eosin staining revealed the infiltration of polymorphonuclears (asterisks) in all SHR groups. WKY—Wistar Kyoto normotensive control rats, SHR—wild type spontaneously hypertensive rats, TH—hyperthyroid rats, HY—hypothyroid rats and SHR^M^—hairless spontaneously hypertensive rats, n = 5 per group. Scale bar represents 200 µm.

**Figure 2 biomedicines-10-01707-f002:**
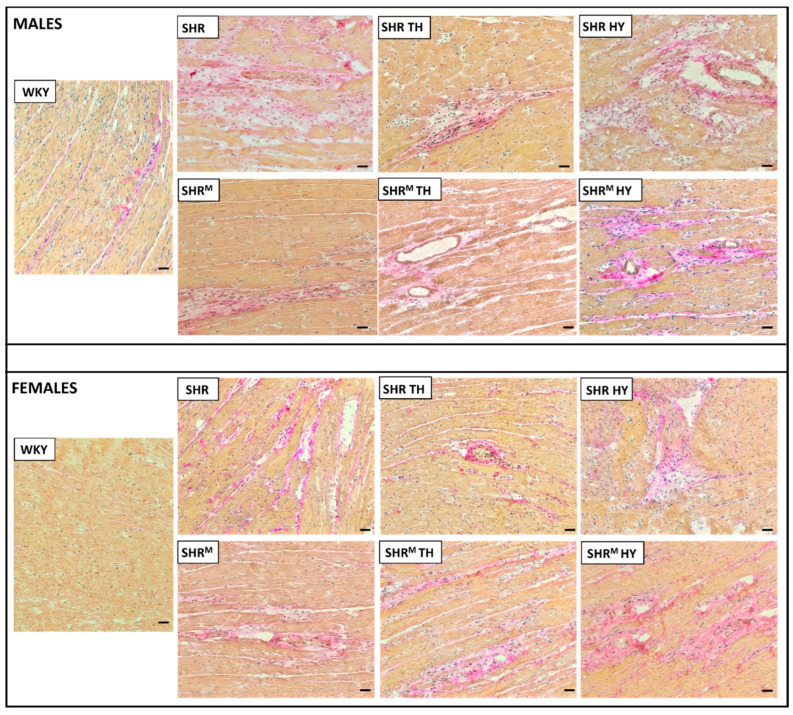
Van Gieson staining shows the collagen deposition (pink) of various degrees in all SHR groups with massive patchy fibrosis in hypothyroid rats. WKY—Wistar Kyoto euthyroid control rats, SHR—wild type spontaneously hypertensive rats, TH—hyperthyroid rats, HY—hypothyroid rats and SHR^M^—hairless spontaneously hypertensive rats, n = 5 per group. Scale bar represents 200 µm.

**Figure 3 biomedicines-10-01707-f003:**
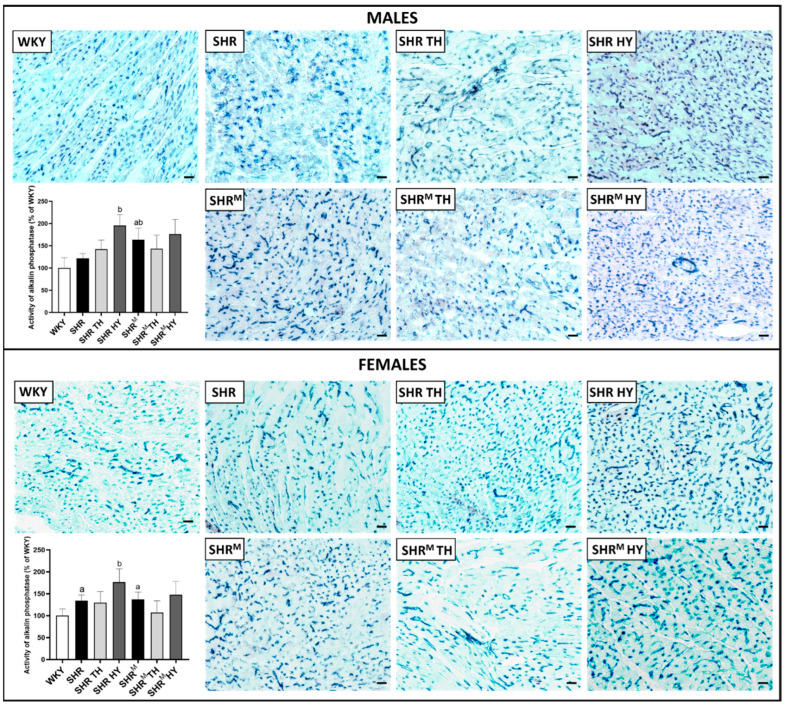
Histochemical demonstration of the alkaline phosphatase (AP) activity (blue) in endothelial cells of the arterial portion of capillaries and quantification of the intensity of the reaction. Note the enhanced AP activity in SHR^M^ males compared to the wild type strain, as well as in response to the hypothyroid status. WKY—Wistar Kyoto normotensive control rats, SHR—wild type spontaneously hypertensive rats, TH—hyperthyroid rats, HY—hypothyroid rats and SHR^M^—hairless spontaneously hypertensive rats, n = 5 per group. Scale bar represents 200 µm. Data are presented as means ± SD, ^a^ *p* < 0.05 vs. WKY and ^b^ *p* < 0.05 vs. SHR. One-way ANOVA and Bonferroni’s multiple comparison test were used as the statistical method.

**Figure 4 biomedicines-10-01707-f004:**
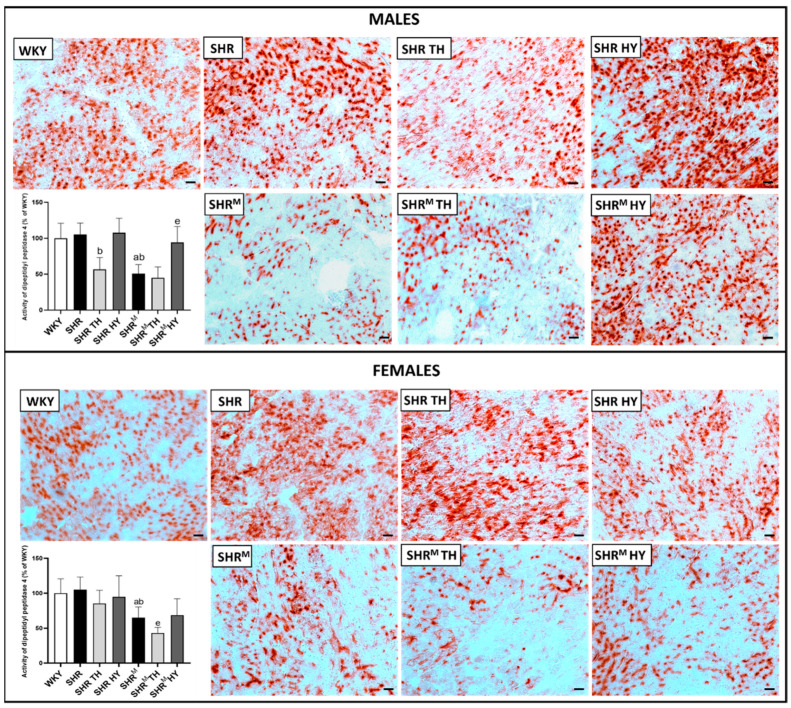
Histochemical demonstration of dipeptidyl peptidase-4 (DPP4) activity (red) in endothelial cells of the venous portion of the capillary network and quantification of the intensity of the reaction. Note the significant decrease of DPP4 activity in SHR^M^ compared to SHR, regardless of the sex, as well as in response to the hyperthyroid status. WKY—Wistar Kyoto normotensive control rats, SHR—wild type spontaneously hypertensive rats, TH—hyperthyroid rats, HY—hypothyroid rats and SHR^M^—hairless spontaneously hypertensive, n = 5 per group. Scale bar represents 200 µm. Data are presented as means ± SD, ^a^ *p* < 0.05 vs. WKY, ^b^ *p* < 0.05 vs. SHR, ^e^ *p* < 0.05 vs. SHR^M^. One-way ANOVA and Bonferroni’s multiple comparison test were used as the statistical method.

**Figure 5 biomedicines-10-01707-f005:**
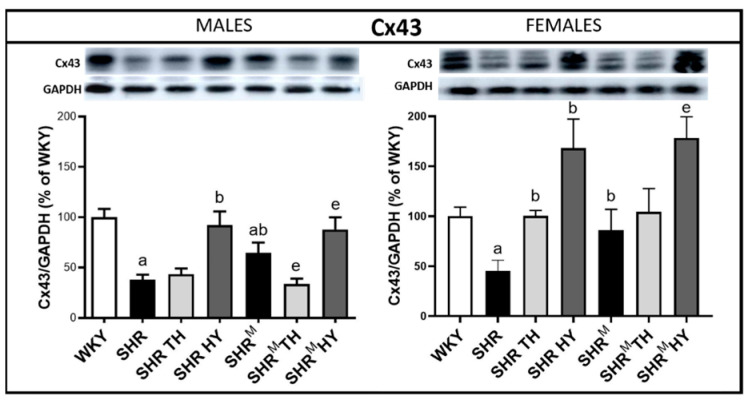
Myocardial protein levels of Cx43 assessed by Western blots. Note the significant decrease of the Cx43 levels in the hypertensive strains compared to normotensive WKY but, to a lesser extent, in hairless SHR^M^ males and females. The hypothyroid status enhanced the Cx43 protein levels in both SHR strains. WKY—Wistar Kyoto normotensive control rats, SHR—wild type spontaneously hypertensive rats, TH—hyperthyroid rats, HY—hypothyroid rats and SHR^M^—hairless spontaneously hypertensive rats, n = 5 per group. Data are presented as means ± SD, ^a^ *p* < 0.05 vs. WKY, ^b^ *p* < 0.05 vs. SHR, ^e^ *p* < 0.05 vs. SHR^M^. One-way ANOVA and Bonferroni’s multiple comparison test were used as the statistical method.

**Figure 6 biomedicines-10-01707-f006:**
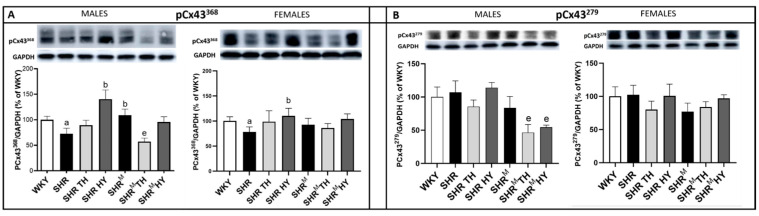
Protein levels of Cx43 variants pCx43^368^ (**A**) and pCx43^279^ (**B**) assessed by the Western blot analysis. Note the reduced levels of pCx43^368^ in wild type SHR males compared to WKY but not in hairless SHR^M^ males. WKY—Wistar Kyoto normotensive control rats, SHR—wild type spontaneously hypertensive rats, TH—hyperthyroid rats, HY—hypothyroid rats and SHR^M^—hairless spontaneously hypertensive rats, n = 5 in each group. Data are presented as means ± SD, ^a^ *p* < 0.05 vs. WKY, ^b^ *p* < 0.05 vs. SHR and ^e^ *p* < 0.05 vs. SHR^M^. One-way ANOVA and Bonferroni’s multiple comparison test were used as the statistical method.

**Figure 7 biomedicines-10-01707-f007:**
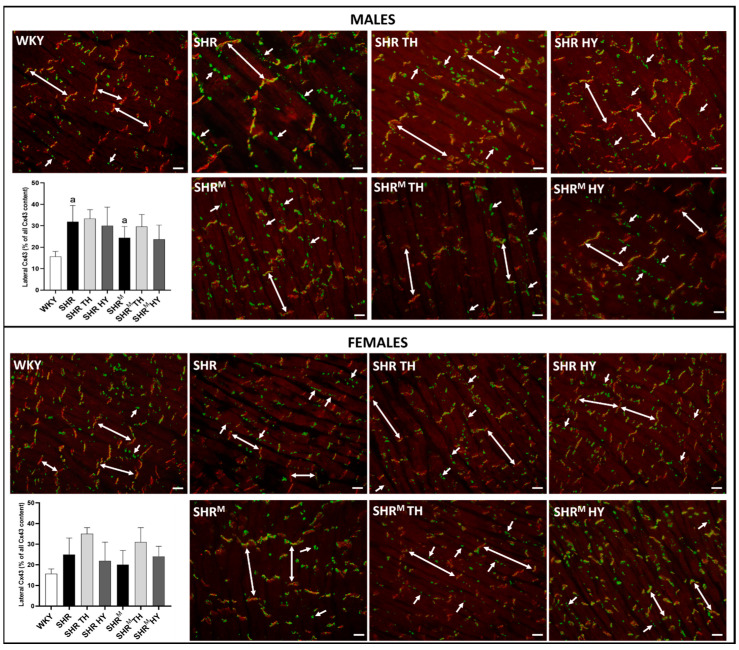
Detection of the myocardial topology of Cx43 using immunofluorescence labeling. Double arrows represent the colocalization of Cx43 (green) with cadherin (red) at the intercalated discs of the cardiomyocytes, and simple arrows show Cx43 localization on the lateral sides of the cardiomyocytes. The graphs represent the quantification of lateral Cx43. Note the abnormal lateral localization is suppressed in male SHR^M^ compared to SHR. WKY—Wistar Kyoto normotensive control rats, SHR—wild type spontaneously hypertensive rats, TH—hyperthyroid rats, HY—hypothyroid rats and SHR^M^—hairless spontaneously hypertensive rats, n = 5 per group. Scale bar represents 400 µm. Data are presented as means ± SD, ^a^ *p* < 0.05 vs. WKY. One-way ANOVA and Bonferroni’s multiple comparison test were used as the statistical method.

**Figure 8 biomedicines-10-01707-f008:**
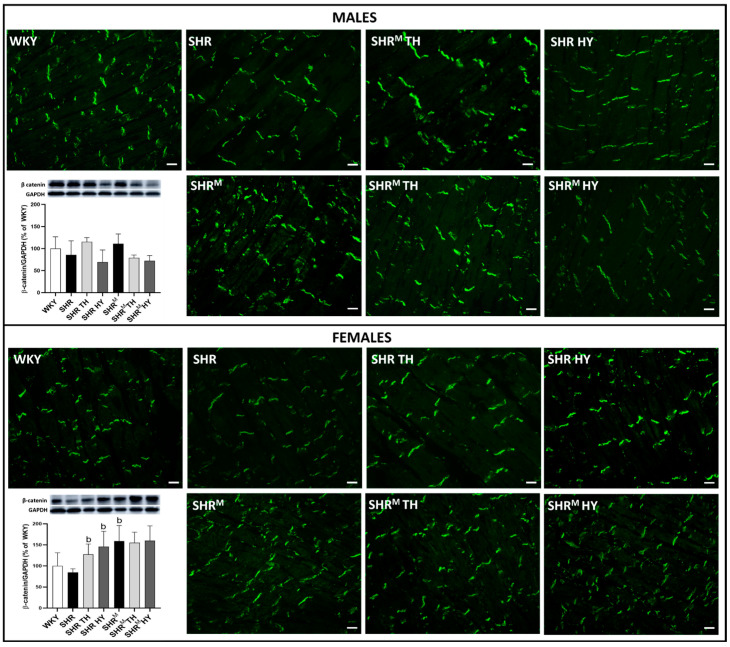
Detection of the myocardial topology in β-catenin (green) using immunofluorescence labeling. Graphs represent quantification of protein levels of β-catenin determined by Western blot analysis. Note the enhanced β-catenin in hairless SHR^M^ males and, to a lesser extent, in females compared to the wild type strain. WKY—Wistar Kyoto normotensive control rats, SHR—wild type spontaneously hypertensive rats, TH—hyperthyroid rats, HY—hypothyroid rats, SHR^M^—hairless spontaneously hypertensive rats, n = 5 per group. Scale bar represents 400 µm. Data are presented as means ± SD, ^b^ *p* < 0.05 vs. SHR. One-way ANOVA and Bonferroni’s multiple comparison test were used as the statistical method.

**Figure 9 biomedicines-10-01707-f009:**
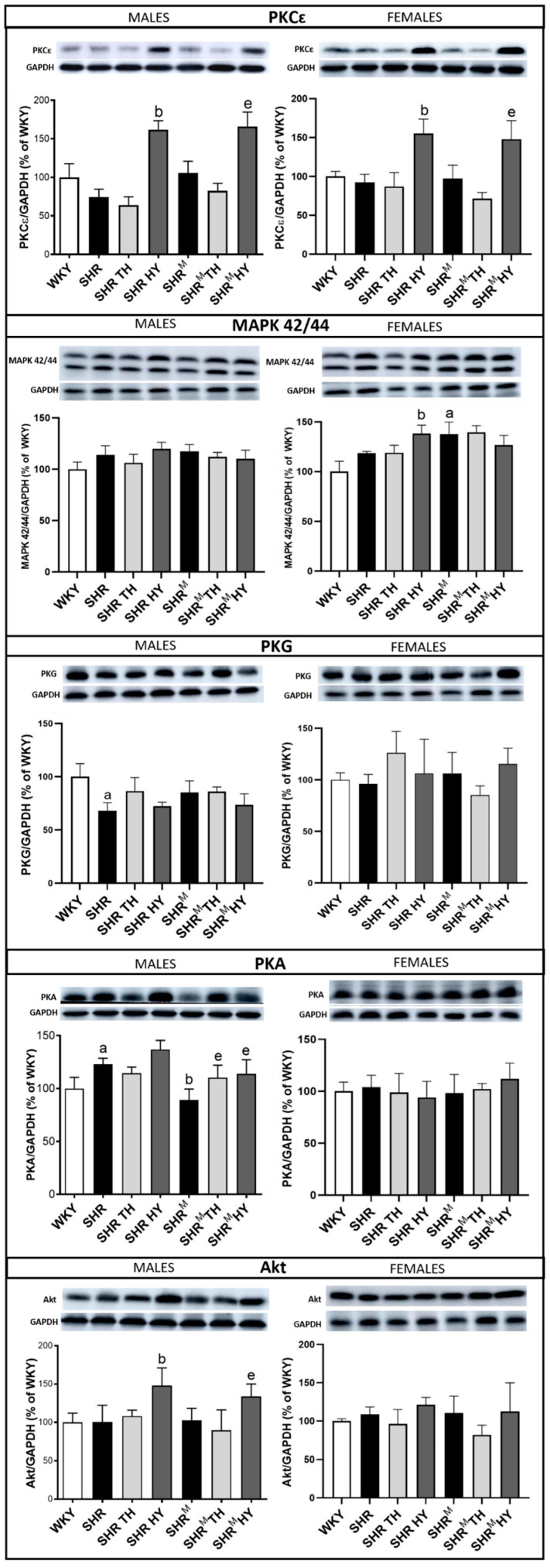
Myocardial levels of the Cx43 interacting protein kinases assessed by Western blotting. Note an increase of PKCε in hairless SHR^M^ males compared to the wild type strain, as well as in response to a hypothyroid status, regardless of the strain and sex. WKY—Wistar Kyoto normotensive control rats, SHR—wild type spontaneously hypertensive rats, TH—hyperthyroid rats, HY—hypothyroid rats and SHR^M^—hairless spontaneously hypertensive rats, n = 5 per group. Data are presented as means ± SD, ^a^ *p* < 0.05 vs. WKY, ^b^ *p* < 0.05 vs. SHR and ^e^ *p* < 0.05 vs. SHR^M^. One-way ANOVA and Bonferroni’s multiple comparison test were used as the statistical method.

**Figure 10 biomedicines-10-01707-f010:**
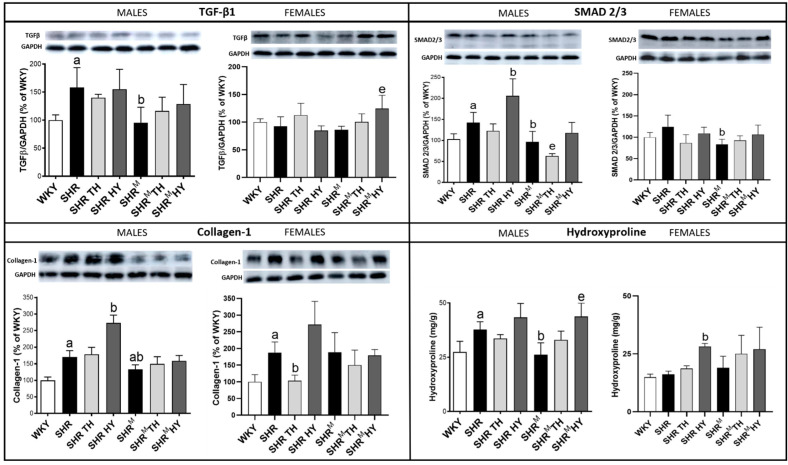
Myocardial protein levels of profibrotic markers TGF-β1, SMAD2/3 and Collagen-1 determined by Western blotting, as well as the levels of hydroxyproline. Note the reduced levels of these proteins in hairless SHR^M^ compared to wild type SHR, apparently in males. WKY—Wistar Kyoto normotensive control rats, SHR—wild type spontaneously hypertensive rats, TH—hyperthyroid rats, HY—hypothyroid rats and SHR^M^—hairless spontaneously hypertensive rats, n = 5 per group. Data are presented as means ± SD, ^a^ *p* < 0.05 vs. WKY, ^b^ *p* < 0.05 vs. SHR and ^e^ *p* < 0.05 vs. SHR^M^. One-way ANOVA and Bonferroni’s multiple comparison tests were used as the statistical method.

**Figure 11 biomedicines-10-01707-f011:**
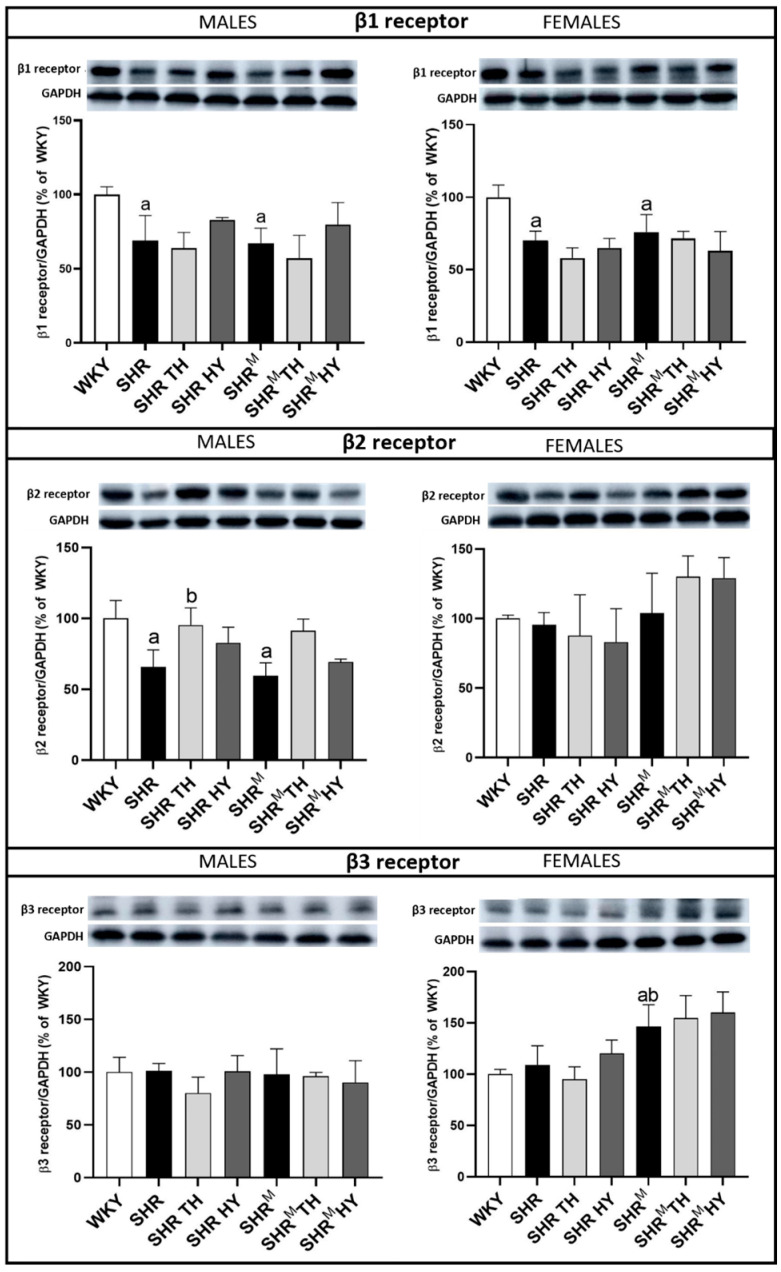
Myocardial protein levels of the β-adrenergic receptors (AR) determined by Western blotting. Note the reduced protein levels of β1-AR and β2-AR in both the SHR and SHR^M^ strains, compared to normotensive WKY rats, as well as an increase of the β3-AR protein in hairless SHR^M^ females compared to wild type SHR. WKY—Wistar Kyoto normotensive control rats, SHR—wild type spontaneously hypertensive rats, TH—hyperthyroid rats, HY—hypothyroid rats and SHR^M^—hairless spontaneously hypertensive rats, n = 5 per group. Data are presented as means ± SD, ^a^ *p* < 0.05 vs. WKY, ^b^ *p* < 0.05 vs. SHR. One-way ANOVA and Bonferroni’s multiple comparison test were used as the statistical method.

**Table 1 biomedicines-10-01707-t001:** List of primary antibodies used for the Western blot detection of proteins.

Protein	Dilution	Product Number	Manufacturer of Antibody
Cx43	1:5000	C6219	Sigma-Aldrich, Missouri, USA
pCx43^368^	1:1000	sc-101660	Santa Cruz Biotechnology, Texas, USA
pCx43^279^	1:1000	sc-12900	Santa Cruz Biotechnology, Texas, USA
β-catenin	1:1000	sc-7963	Santa Cruz Biotechnology, Texas, USA
PKCε	1:1000	sc-214	Santa Cruz Biotechnology, Texas, USA
TGF-β1	1:1000	SAB4502954	Sigma-Aldrich, Missouri, USA
SMAD2/3	1:1000	#3102	Cell Signaling Technology, Colorado, USA
Collagen-1	1:1000	ab90395	Abcam, United Kingdom
PKA	1:1000	sc-365615	Santa Cruz Biotechnology, Texas, USA
PKG	1:1000	sc-25429	Santa Cruz Biotechnology, Texas, USA
AktK	1:1000	sc-81436	Santa Cruz Biotechnology, Texas, USA
MAPK42/44	1:1000	137F5	Cell Signaling Technology, Colorado, USA
β1AR	1:1000	ab3442	Abcam, Cambridge, United Kingdom
β2AR	1:1000	bs-0947R	Bioss, Woburn, Massachusetts, USA
β3AR	1:1000	bs-1063R	Bioss, Massachusetts, USA
GAPDH	1:1000	sc-25778	Santa Cruz Biotechnology, Texas, USA

**Table 2 biomedicines-10-01707-t002:** General characteristics of the experimental rats.

**MALES—Variables**	**WKY**	**SHR**	**SHR TH**	**SHR HY**	**SHR^M^**	**SHR^M^ TH**	**SHR^M^ HY**	
**BW (g)**	410 ± 1.41	398.33 ± 21.08	286 ± 1.41 ^b^	383.5 ± 7.78	359 ± 42.76	307.33 ± 31.37	365.25 ± 38.46	
**HW (g)**	1.05 ± 0.01	1.53 ± 0.08 ^a^	1.43 ± 0.05	0.87 ± 0.21 ^b^	1.59 ± 0.13 ^a^	2.01 ± 0.14 ^e^	1.14 ± 0.25 ^e^	
**LVW (g)**	0.75 ± 0.01	1.19 ± 0.08 ^a^	1.06 ± 0.05	0.71 ± 0.08 ^b^	1.21 ± 0.13 ^a^	1.46 ± 0.11	0.89 ± 0.18 ^e^	
**TBARS (nmol/mg)**	1.75 ± 0.29	1.66 ± 0.45	2.06 ± 0.11	1.99 ± 0.21	2.00 ± 0.10	2.37 ± 0.45	1.75 ± 0.22	
**GSH (µmol/g)**	1.99 ± 1.13	1.97 ± 0.27	2.06 ± 0.33	2.43 ± 0.08	2.14 ± 0.15	2.27 ± 0.24	2.39 ± 0.11	
**T3 (nmol/L)**	0.84 ± 0.04	1.12 ± 0.02 ^a^	2.84 ± 0.11 ^b^	0.48 ± 0.06 ^b^	1.08 ± 0.03 ^a^	4.05 ± 0.16 ^e^	0.59 ± 0.01 ^e^	
**T4 (nmol/L)**	55.0 ± 0.8	44.0 ± 6.9 ^a^	62.3 ± 1.7 ^b^	12.4 ± 1.9 ^b^	46.2 ± 2.2 ^a^	69.0 ± 1.0 ^e^	16.1 ± 0.7 ^e^	
**TG (mmol/L)**	1.60 ± 0.20	0.66 ± 0.04 ^a^	0.96 ± 0.02 ^b^	0.43 ± 0.03	0.70 ± 0.13 ^a^	0.80 ± 0.06	0.51 ± 0.08	
**TC (mmol/L)**	1.88 ± 0.10	1.12 ± 0.02 ^a^	1.70 ± 0.24 ^b^	1.98 ± 0.07 ^b^	1.18 ± 0.25 ^a^	1.40 ± 0.06	2.42 ± 0.34 ^e^	
**HDL (mmol/L)**	1.33 ± 0.03	0.81 ± 0.04 ^a^	1.38 ± 0.23 ^b^	1.40 ± 0.05 ^b^	0.83 ± 0.14 ^a^	1.12 ± 0.06	1.77 ± 0.33 ^e^	
**LDL (mmol/L)**	0.24 ± 0.09	0.17 ± 0.03	0.15 ± 0.04	0.53 ± 0.10	0.20 ± 0.10	0.13 ± 0.04	0.58 ± 0.21 ^e^	
**TC/HDL (ratio)**	1.43 ± 0.10	1.39 ± 0.04	1.24 ± 0.03	1.40 ± 0.05	1.41 ± 0.09	1.26 ± 0.04	1.39 ± 0.20	
**FEMALES—Variables**	**WKY**	**SHR**	**SHR TH**	**SHR HY**	**SHR^M^**	**SHR^M^ TH**	**SHR^M^ HY**	
**BW (g)**	241.41 ± 2.83	201.50 ± 1.26	197.21 ± 29.70	200.67 ± 1.15	210.60 ± 13.67 ^a^	212.50 ± 9.19	185.50 ± 33.74	
**HW (g)**	0.752 ± 0.07	0.827 ± 0.06 ^a^	1.116 ± 0.25 ^b^	0.679 ± 0.05	0.917 ± 0.10 ^a^	1.101 ± 0.25	0.802 ± 0.14	
**LVW (g)**	0.544 ± 0.04	0.639 ± 0.02 ^a^	0.884 ± 0.22 ^b^	0.512 ± 0.01 ^b^	0.624 ± 0.23 ^a^	0.737 ± 0.12	0.540 ± 0.16	
**TBARS (nmol/mg)**	2.25 ± 0.27	2.33 ± 0.25	2.90 ± 0.07	2.85 ± 0.41	2.56 ± 0.43	3.51 ± 0.35 ^e^	3.85 ± 0.27	
**GSH (µmol/g)**	1.87 ± 0.07	1.98 ± 0.01	1.88 ± 0.18	2.21 ± 0.46	1.98 ± 0.15	1.97 ± 0.19	2.40 ± 0.31	
**T3 (nmol/L)**	0.92 ± 0.02	1.29 ± 0.02 ^a^	2.83 ± 0.11	0.38 ± 0.06 ^b^	1.08 ± 0.04 ^b^	2.75 ± 0.05 ^e^	0.51 ± 0.02 ^e^	
**T4 (nmol/L)**	47.6 ± 2.1	42.9 ± 1.8 ^a^	58.3 ± 1.0^b^	14.8 ± 1.5 ^b^	43.8 ± 0.7 ^a^	62.7 ± 0.9 ^e^	18.2 ± 0.3 ^e^	
**TG (mmol/L)**	1.07 ± 0.26	1.28 ± 0.02	1.05 ± 0.34	0.56 ± 0.05 ^b^	1.03 ± 0.19	1.16 ± 0.14	0.39 ± 0.06 ^e^	
**TC (mmol/L)**	2.28 ± 0.32	1.61 ± 0.04 ^a^	1.86 ± 0.21	1.40 ± 0.10	1.37 ± 0.06 ^a^	1.59 ± 0.07	2.70 ± 0.07 ^e^	
**HDL (mmol/L)**	1.85 ± 0.31	1.19 ± 0.02 ^a^	1.47 ± 0.36	1.01 ± 0.16	1.07 ± 0.08 ^a^	1.31 ± 0.05	2.07 ± 0.17 ^e^	
**LDL (mmol/L)**	0.21 ± 0.07	0.17 ± 0.02	0.14 ± 0.04	0.28 ± 0.7	0.11 ± 0.10	0.11 ± 0.06	0.63 ± 0.04 ^e^	
**TC/HDL (ratio)**	1.26 ± 0.05	1.36 ± 0.02	1.28 ± 0.14	1.41 ± 0.13	1.29 ± 0.14	1.24 ± 0.09	1.36 ± 0.05	

BW—body weight, HW—heart weight, LVW—left ventricular weight, TBARS—thiobarbituric acid reactive substances in the left ventricle, GSH—reduced glutathione in the left ventricle, T_3_—total triiodothyronine, T_4_—total thyroxine, TG—plasma triglycerides, TC—total cholesterol, HDL—high-density lipoprotein-cholesterol, LDL—low-density lipoprotein-cholesterol, WKY—Wistar Kyoto normotensive control rats, SHR—wild type spontaneously hypertensive rats, TH—hyperthyroid rats, HY—hypothyroid rats, SHR^M^—hairless spontaneously hypertensive rats, n = 5 per group. Data are presented as means ± SD, ^a^ *p* < 0.05 vs. WKY, ^b^ *p* < 0.05 vs. SHR and ^e^ *p* < 0.05 vs. SHR^M^. One-way ANOVA and Bonferroni’s multiple comparison tests were used as the statistical method.

## Data Availability

Not applicable.

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
