# Peer review of "Cardiac Cx43 Signaling Is Enhanced and TGF-β1/SMAD2/3 Suppressed in Response to Cold Acclimation and Modulated by Thyroid Status in Hairless SHR^M^"

_biomedicines, 2022, doi:10.3390/biomedicines10071707_

Round 1

Reviewer 1 Report

The manuscript by Andelova et al. defines the role of a gap junction related myocardial Cx43 protein in response to cold impacts and to thyroid status, by utilizing spontaneous hypertensive rats and its mutant variant. The authors further revealed that mutant SHR might be less prone to developing malignant arrhythmias then wild type SHR, due to an increased alkaline phosphatase activity, and a decreased dipeptidyl peptidase-4 activity in both males and females. While SHR hearts showed reduced Cx43 signalling, this was not the case for the mutant hearts. In addition, markers of pro-fibrotic pathways TGF-β1 and SMAD2/3 were lower in the SHR male mutant hearts.  The study is interesting and demonstrates gender specific responses, although a few concerns need to be addressed.

Major comments:

 It would be indeed interesting to see some functional readouts namely, cardiac function and/or electrophysiology of the heart in response to cold/thyroid status in SHR and SHR mutant. Additionally, the authors state in line 457, “Besides the apparent hypertrophy….”. Did the authors measure cardiac hypertrophy? The manuscript does not show the levels of hypertrophy markers (Nppa. Nppb etc) or any other indices of hypertrophy, in the SHR or AHR mutants.

Figures 1 and 2: In addition to the staining images, a bar graph showing the quantification of infiltration of polymorphonuclears and fibrosis, and statistics between the different groups would be helpful.

Is there any cardiomyocyte apoptosis observed in the SHR and does SHR mutant hearts exhibit less apoptosis? Blotting for some apoptotic markers or staining the heart sections with TUNEL staining could be done.

The possible causes for the differences observed between the males and females SHR mutants and wild-type hearts should be elaborately discussed.

Minor comments:

Please define SHR in the first sentence of the abstract.

It would be helpful to describe the rationale for performing the alkaline phosphatase (AP) and dipeptidyl peptidase-4 (DPP4) activities in the context of fibrosis in the result section. For, eg. a sentence mentioning what happens when AP/DPP4 activity is increased or decreased can be added before presenting the results.

Author Response

Reviewer 1

First of all, we would like to thank very much for reviewing our manuscript. We appreciate relevant remarks and suggestions aimed to increase quality of current study and our research. Unfortunately, we were not able to respond adequately to all issues as you can see below. However, key findings of this “pilot” study encourage us to perform further investigations including these you suggested.

The manuscript by Andelova et al. defines the role of a gap junction related myocardial Cx43 protein in response to cold impacts and to thyroid status, by utilizing spontaneous hypertensive rats and its mutant variant. The authors further revealed that mutant SHR might be less prone to developing malignant arrhythmias then wild type SHR, due to an increased alkaline phosphatase activity, and a decreased dipeptidyl peptidase-4 activity in both males and females. While SHR hearts showed reduced Cx43 signalling, this was not the case for the mutant hearts. In addition, markers of pro-fibrotic pathways TGF-β1 and SMAD2/3 were lower in the SHR male mutant hearts.  The study is interesting and demonstrates gender specific responses, although a few concerns need to be addressed.

Major comments:

 It would be indeed interesting to see some functional readouts namely, cardiac function and/or electrophysiology of the heart in response to cold/thyroid status in SHR and SHR mutant.

Yes, we fully agree with you that assessment of the cardiac function and/or electrophysiology may provide answer as whether SHR mutants benefit from enhanced Cx43 signalling and supressed ECM signalling.  We have also noted this fact as a limitation of the study. Indeed, we were limited by number of hairless SHR mutants, which are not commercially available. Therefore, it was not real to perform additional experiments to test the propensity of the heart to arrhythmias as well as heart function. We would like to do it in further study as we noted in the paragraph of Limitation. (But we need to wait for 9-month-old animals). 

Additionally, the authors state in line 457, “Besides the apparent hypertrophy….”. Did the authors measure cardiac hypertrophy? The manuscript does not show the levels of hypertrophy markers (Nppa. Nppb etc) or any other indices of hypertrophy, in the SHR or SHR mutants.

Both rat strains, i.e. SHR and hairless SHR mutants represent animal model of essential (primary) hypertension due to pressure overload. There was no difference in elevated blood pressure between these strains (wild type SHR 190.0 ± 7 mmHg in males and 179.6 ± 12 mmHg in females, hairless SHRM 182.9 ± 6 mmHg in males and 169.7 ± 9 mmHg in females). SHR strain is characterized by hypertrophy and fibrosis over time (as it was published by several our previous (see f.e. Radosinska et al. 2013) and other studies (Bing et al. 1995). Considering these facts we confirmed (“apparent”) hypertrophy according to increased left ventricular weight (SHR: 1.19 g males and 0.639 g females; hairless SHR 1.21g males and 0.624 females; versus WKY rats: 0.75 males and 0.544 females) and visualy by histological images of hypertrophied cardiomyocytes. We did not detect hypertrophy markers like Nppa or Nppb but we agree with you that perhaps these could reveal gentle differences in the hypertrophy status between both SHR strains. We hope to examin it in future experiments.

Figures 1 and 2: In addition to the staining images, a bar graph showing the quantification of infiltration of polymorphonuclears and fibrosis, and statistics between the different groups would be helpful.

Yes, it would be perfect to quantify polymorphonuclears infiltration status. However, we did not find any paper in which it was done in addition to the HE staining. Maybe because it is not possible to differentiate stained nuclei in polymorphonulears from other stained nuclei in the myocardial tissue, i.e. those in cardiomyocytes, vessels, and extracellular cells. Thus it is hard to perform quantitative image analysis via extracting blue colour of nuclei only in polymorphonuclears.

Regarding fibrosis, we would like to emphasize that by representative microscopic images in Figure 2 we wanted to show the pattern/type of fibrosis, i.e. that it is patchy. Moreover, fibrosis was heterogeneously distributed within the left ventricle.Due to this fact, we quantified fibrosis in the whole left ventricle homogenate via determination of collagen content by Western blot and measurement of hydroxyproline using spectrophotometric method.

Is there any cardiomyocyte apoptosis observed in the SHR and does SHR mutant hearts exhibit less apoptosis? Blotting for some apoptotic markers or staining the heart sections with TUNEL staining could be done.

It was reported 2.6-fold increase of apoptosis in the heart of 16-weeks-old SHR relative to WKY controls (Hamet 1995). Moreover, plasma apoptotic markers were increased in patient suffering from essential hypertension (Morillas 2022). However, because we focused in our study mainly on markers associated with Cx43 mediated intercellular communication we did not examine the markers of cardiomyocyte death. Nevertheless, we agree with you that it would be interesting to consider the impact of apoptosis in hairless SHR strain versus wild type SHR in the context of heart function. This could be done in further study since we have used most of our heart samples.

The possible causes for the differences observed between the males and females SHR mutants and wild-type hearts should be elaborately discussed.

There is a gap of knowledge and we think that increased thermogenesis induced by cold might be possible cause for observed cardiac differences in the SHRM versus wild type SHR. Findings challenge to explore mechanisms by which adaptation to cold triggers cardiac responses. We noted this in the paragraph of Conclusion.

Minor comments:

Please define SHR in the first sentence of the abstract.

Yes, it was done.

It would be helpful to describe the rationale for performing the alkaline phosphatase (AP) and dipeptidyl peptidase-4 (DPP4) activities in the context of fibrosis in the result section. For, eg. a sentence mentioning what happens when AP/DPP4 activity is increased or decreased can be added before presenting the results.

According to your suggestion, we added in the text: Alterations of capillary network AP activity may reflect myocardial adaptation to hypertension to maintain function of structurally remodelled heart. Enhanced DPP4 activity is considered as detrimental in pathophysiological conditions due to implication in collagen metabolism and pro-inflammatory signalling.

Submission Date

06 June 2022

Date of this review

19 Jun 2022 18:34:06

Reviewer 2 Report

In their manuscript entitled "Cardiac Cx43 signalling is enhanced and TGF-β1/SMAD2/3 suppressed in response to cold acclimation and modulated by thyroid status in hairless SHRM" the author's present data that suggests cold acclimation or alternatively thyroid hormone modulation may represent elements of arrhythmias that could be exploited therapeutically and warrant further investigation. The article overall is well organized and written, and figures are clear and easy to understand.  There are a number of concerns that I have outlined below that should be addressed before publication. 

Line 47: should read --> the heart is an electro-mechanic pump

Lines 109-110: Should read --> the chest was opened

Section 2.6: 1) inconsistent use of anti- for the antibodies

2) this section could be reorganized to be smaller by binning antibodies by manufacturer such as “Antibodies purchased from Santa Cruz (Santa Cruz Biotechnology, Dallas, TX, USA) were X (dilution, product number), Y (dilution, product number) etc….

    Alternatively this could be easily organized into a table 

        3) While GAPDH is a classically acceptable loading and normalization control for heart tissue, total protein stains are becoming the preferred loading control and normalizing factor.

Figure 5: For females it is clear that Cx43 is migrating in the P0, P1, P2 migratory bands, this is less evident in the males in the included image.  When looking at the provided original images I am curious if this blot was mixed up with the corresponding GADPH which for males shows three bands of reactivity, but females only one band?  New supplemental images of the complete membrane showing full ladder could resolve this.

Section 3.6: Although the results support the notion that kinase expression is changed and that may contribute to altered regulation of Cx43, it is a long jump to make that conclusion without testing the expression of the activated forms of these kinases (e.g.  PKCε pS729 normalized to total PKCε).  This is essential to properly interpret the change in kinase expression and potential impact on Cx43 phosphorylation and function.  

Line 314 and Line 478: Are apparently contradictory, in 314 the implication is pS279 facilitates channel conductivity (enhancing), and in 478 is said to hamper GJIC (decreasing), this should be clarified.  This is one of the two MAPK phosphorylation sites on Cx43 (the other pS282) and when phosphorylated lead to channel closure and prime Cx43 for interactions with Nedd4 leading to ubiquitination, internalization, and degradation. 

Section 3.4 and Figure 7 are particularly interesting, as it appears that some of the lateralized Cx43 remains in dense staining plaque like structures. Our lab has observed this population of Cx43 staining becoming diffuse and less intense suggesting disruption of the gap junction plaques. I would like to see triton X-100 solubility assay results either in situ or in vitro following protein isolation (in the absence of ionic detergent or reducing agents) to determine what percentage of Cx43 is remaining in gap junction plaques as well as its phosphorylation pattern. 

I’m curious about rationale for choosing for Bonferroni’s multiple comparison test and wonder if Sidak’s test is more appropriate given the dataset. 

Line 453: “While the latter deteriorate Cx43 channel function [31]” this sentence seems incomplete, at a minimum difficult to understand “which latter” is being referred to. Suggest rewording or combining with the previous sentence.

Sections 3.5 and 3.7: Similar to the point made regarding kinase activity in Section 3.6 it would be useful to have information regarding the active vs. inactive forms of beta-catenin, SMAD2/3, and TGF-beta (requires non-reducing conditions). 

In figure legends it would be beneficial to the reader to include the statistical method used in the figure, this will aid them in being "stand alone".

Author Response

Reviewer 2

First of all, we would like to thank you very much for reviewing our manuscript. We recognized you as an expert in Cx43 research and appreciate your criticism, suggestion and remarks. It is challenging to pay attention in our further research. Although we would like to respond all issues you addressed adequately and sufficiently, in fact, our responses reflect our possibilities, conditions and limitations. Nevertheless, key findings of this “pilot” study along with your fundamental remarks encourage us to perform further study and investigations including these you suggested.

In their manuscript entitled "Cardiac Cx43 signalling is enhanced and TGF-β1/SMAD2/3 suppressed in response to cold acclimation and modulated by thyroid status in hairless SHRM" the author's present data that suggests cold acclimation or alternatively thyroid hormone modulation may represent elements of arrhythmias that could be exploited therapeutically and warrant further investigation. The article overall is well organized and written, and figures are clear and easy to understand.  There are a number of concerns that I have outlined below that should be addressed before publication. 

Line 47: should read --> the heart is an electro-mechanic pump

Lines 109-110: Should read --> the chest was opened

Thank you, we corrected mistakes.

Section 2.6: 1) inconsistent use of anti- for the antibodies

2) this section could be reorganized to be smaller by binning antibodies by manufacturer such as “Antibodies purchased from Santa Cruz (Santa Cruz Biotechnology, Dallas, TX, USA) were X (dilution, product number), Y (dilution, product number) etc….

    Alternatively this could be easily organized into a table 

We followed your suggestion and included Table 1.

        3) While GAPDH is a classically acceptable loading and normalization control for heart tissue, total protein stains are becoming the preferred loading control and normalizing factor.

Thanks for your information. We agree with you and we would like to use this approach in the future.

Figure 5: For females it is clear that Cx43 is migrating in the P0, P1, P2 migratory bands, this is less evident in the males in the included image.  When looking at the provided original images I am curious if this blot was mixed up with the corresponding GADPH which for males shows three bands of reactivity, but females only one band?  New supplemental images of the complete membrane showing full ladder could resolve this.

Unfortunately, P0, P1, P2 migrating Cx43 forms are not clearly visible in males in Figure 5. However, as you can see in supplementary WB images the obvious three bands could be recognized. Three bands in males SHR are also clearly recognized in our previous studies (e.g. Radosinska 2013).

However, we think that GAPDH did not interfere with Cx43 band in males. In some supplementary original images is possible to see all bands of connexin 43 that were stripped by stripping buffer before GAPDH exposure. Because the bands were not removed adequately, so we decided not to strip membranes but to use another membrane for detection of GAPDH in females.

We apologize that we cannot provide GAPDH with protein ladder as we did not use protein marker at that membrane.

Section 3.6: Although the results support the notion that kinase expression is changed and that may contribute to altered regulation of Cx43, it is a long jump to make that conclusion without testing the expression of the activated forms of these kinases (e.g.  PKCε pS729 normalized to total PKCε).  This is essential to properly interpret the change in kinase expression and potential impact on Cx43 phosphorylation and function.

Yes, we fully agree with you and we added in the text: Demonstrated changes of protein kinases expression may have impact on Cx43 phosphorylation and function. Therefore, their interaction with Cx43 (co-immunoprecipitation) along with its phosphorylated status is challenging to explore.

Line 314 and Line 478: Are apparently contradictory, in 314 the implication is pS279 facilitates channel conductivity (enhancing), and in 478 is said to hamper GJIC (decreasing), this should be clarified.  This is one of the two MAPK phosphorylation sites on Cx43 (the other pS282) and when phosphorylated lead to channel closure and prime Cx43 for interactions with Nedd4 leading to ubiquitination, internalization, and degradation. 

Sorry for our mistake, pS279 hamper/decrease channel conductivity as we corrected in the text.

Section 3.4 and Figure 7 are particularly interesting, as it appears that some of the lateralized Cx43 remains in dense staining plaque like structures. Our lab has observed this population of Cx43 staining becoming diffuse and less intense suggesting disruption of the gap junction plaques. I would like to see triton X-100 solubility assay results either in situ or in vitro following protein isolation (in the absence of ionic detergent or reducing agents) to determine what percentage of Cx43 is remaining in gap junction plaques as well as its phosphorylation pattern. 

According to our experience using transmission electron microscopy the lateral localisation of Cx43 is mostly confined to GJ plague in healthy heart (see f.e Andelova 2021, Fig 1c and Fig. 4b,d) as well as in various cardiac pathologies including hypertensive rats (see e.g. Radosinska 2013, Fialova 2008). These lateral plaques frequently underwent internalization followed by degradation as demonstrated in cited papers. However, as we understood you have observation suggesting plaque disruption prior its internalisation. It is very interesting and attractive view.  However, because our limitations regarding the samples we are not able to perform triton X-100 solubility test now only in further study. Concern the importance of Triton X-100 soluble and Triton X-100 insoluble portions analysis we included reference (Trease 2017) in revised text

I’m curious about rationale for choosing for Bonferroni’s multiple comparison test and wonder if Sidak’s test is more appropriate given the dataset. 

Considering that Bonferoni’s and Sidak’s methods are very similar and both are used by researchers, including our group. Though Sidak’s method is bit more powerful than Bonferoni’s we will use it in our further studies.

Line 453: “While the latter deteriorate Cx43 channel function [31]” this sentence seems incomplete, at a minimum difficult to understand “which latter” is being referred to. Suggest rewording or combining with the previous sentence.

We revised this sentence to express clearly that inflammation deteriorates Cx43 channel function.

Sections 3.5 and 3.7: Similar to the point made regarding kinase activity in Section 3.6 it would be useful to have information regarding the active vs. inactive forms of beta-catenin, SMAD2/3, and TGF-beta (requires non-reducing conditions). 

Regarding active and inactive forms of mentioned markers we would like to explain that with exception of collagen-1 (where in datasheet it was exactly stated to perform WB under non-reducing conditions) we accomplished all protein measurements under reducing conditions, i.e. we detected inactive forms. We decided to use reducing condition because we did not obtain appropriate bands in non-reducing conditions with purchased antibodies. We noted these facts in chapter Methods. No doubts that we should think how to analyse active forms.

In figure legends it would be beneficial to the reader to include the statistical method used in the figure, this will aid them in being "stand alone".

Statistical method was added in Figure legends.

Submission Date

06 June 2022

Date of this review

20 Jun 2022 21:45:22

Round 2

Reviewer 1 Report

In this revised version, the authors have satisfactorily addressed the concerns and recommendations. 

Author Response

Dear reviewer,  thank you very much for Your positive response, we really appreciate your previous comments that help us to improve our article.

Reviewer 2 Report

I would like to thank the author's for their prompt response to this reviewer's concerns. From a standpoint of scientific content and the author's responses I have no other concerns and look forward to future reports from this group.

There are a couple minor grammatical changes and typos in the revised manuscript that will need corrected.

Line 302: "pathophysiological conditions due to due to implication in...." the second "due to" should be deleted. 

Line 372: "In the context of lateral Cx43 distribution the use Triton...." should read "In the context of lateral Cx43 distribution the use of Triton...."

Author Response

Dear Reviewer,

we are very happy that our provided revised version was sufficient. We have tried to include all your suggestions what in our opinion significantly improved our article. Thank you very much for your critically point of view.

We corrected grammatical typos in the manuscript.